# Distribution of spontaneous combustion three zones and optimization of nitrogen injection location in the goaf of a fully mechanized top coal caving face

Yun Qi[1,2]*, Wei Wang[1,2]*, Qingjie Qi[2,3], Zhangxuan Ning[1], Youli Yao[1]

1 School of Coal Engineering, Shanxi Datong University, Datong Shanxi, China, 2 College of Safety Science and Engineering, Liaoning Technical University, Fuxin Liaoning, China, 3 China Coal Technology Engineering Group, Emergency Research Institute, Beijing, China

* qiyun_sx@sxdtdx.edu.cn (YQ); wangwei@sxdtdx.edu.cn (WW)

**Data Availability Statement:** All relevant data are within the paper and its Supporting information files.

## Abstract

In order to effectively prevent and control spontaneous combustion of residual coal in the goaf and reduce the waste of nitrogen caused by setting the position of nitrogen injection, 1303 fully mechanized coal caving faces of the Jinniu Mine are studied. By deploying a bundle tube monitoring system in the inlet air side and return air side of the goaf, changes in gas concentration in the goaf are continuously monitored. In addition, the distribution area for spontaneous combustion three-zone in the goaf is divided into heat dissipation zone, oxidized spontaneous combustion zone, and suffocation zone. Simulations from the COMSOL Multiphysics 5.3 software provide insight based on the three zones division standard of spontaneous combustion in the goaf. The gradual deepening of the nitrogen injection position into the goaf affects the lower limit of the oxidized spontaneous combustion zone significantly, but the impact on the upper limit of the oxidized spontaneous combustion zone is not obvious and is negligible. With regard to the width of the oxidized spontaneous combustion zone, it initially decreases followed by a gradual increase. Numerical calculations suggest the optimal nitrogen injection position is 40 m from the roof cutting line, with an oxidized spontaneous combustion zone width of 28 m. Based on the simulation analysis results, nitrogen injection controlling measures have been adopted for spontaneous combustion of residual coal in the goaf of the 1303 fully mechanized coal caving faces, and coal self-ignition in the goaf has been successfully extinguished.

## 1 Introduction

Coal mine fires affect the safety of mines. Among coal mine fires, 85% stem from the spontaneous combustion of coal in the goaf [1–3]. Fully mechanized top caving technology improves production efficiency of mines, but it increases the leakage intensity of goaf and produces excessive residual coal. The increase in the leakage intensity increases the probability of spontaneous combustion of coal in the goaf [4–7]. Injecting nitrogen technology in mining

**Funding:** The research presented in this paper was supported by the Key Laboratory of Mine Thermodynamic disasters and Control of Ministry of Education (Liaoning Technical University), Huludao, China. Thanks to "The National Natural Science Foundation of China (Grant: 51274113; Key special-funded projects of the State key R & D program (2018YFC0807900)" for the financial support. The recipient of these funds is Qi Qingjie. Qi Qingjie is my doctoral supervisor and co-author of this paper.

**Competing interests:** The authors have dwclared that no competing interests exist.

**Abbreviations: CFD**, Computational Fluid Dynamics; **GC-2010**, model of the gas Chromatograph Model.

operations aids in preventing the spontaneous combustion of coal in the goaf and in extinguishing fires that do occur the goaf.

Many studies involve nitrogen injection and fire prevention in the goaf. Krishna *et al.* [8] used CFD technology to study the change of a flow field and designed a nitrogen injection fire-fighting and extinguishing scheme in goafs. Li Zongxiang *et al.* [9] combined the finite element numerical method with computer graphics technology to simulate the change process of the boundary of nitrogen injection control zone with the amount of nitrogen injection. From this model, Li *et al.* obtained the negative exponential relationship between the amount of nitrogen injection and the boundary of the nitrogen injection control area. Gao Ke *et al.* [10] adopted the Fluent numerical method to optimize nitrogen injection. The method identified the required nitrogen injection amount, location of nitrogen injection, and air volume in the goaf area. Zhu Hongqing *et al.* [11] studied the law of spontaneous combustion and temperature rise in the context of nitrogen injection and fire prevention using the Fluent software.

These studies only focused on nitrogen injection in the context of fire prevention principles and the nitrogen injection process. However, they did not consider the effects of coal oxidation and gas emission in the goaf area.

To address these shortcomings, this work investigates the 1303 fully mechanized caving faces of the Jinniu coal mine. Field measurements were conducted to divide the "three-zone" distribution area of spontaneous combustion in the goaf. Using COMSOL Multiphysics 5.3 [12], the simulation can model the dependence between the spontaneous combustion "three-zone" distribution area and the nitrogen injection position via the leakage flow field and oxygen concentration field. The influence of nitrogen injection position on the width of oxidized spontaneous combustion zone was studied to optimize the location of nitrogen injection. Based on the results of the simulation analysis, the proposed nitrogen injection scheme could hold significant benefits industrial settings.

## 2 Analysis of the three zones of spontaneous combustion in goaf

### 2.1 Overview of 1303 fully mechanized caving mining faces

The 1303 fully mechanized coal caving faces in the Jinniu coal mine are in the mining area of the 1030 level 9+10+11# coal seam. The working face starts from the concentrated return airflow tunnel in the east and extends to the boundary of the well field in the west. The face also stretches from the 1305 working faces in the south to the 1301 working faces in the north. The design length of working faces is 906.6 m, the length of the slope is 90 m, the coal seam thickness is 5.24–7.30 m, and the average thickness is 6.17 m. Adopting fully mechanized top coal caving mining technology, all the caving methods are used to control the roof of the goaf. The mining height of the working face is 2.9 m, the top coal caving height is 3.3 m, and the ratio of mining and discharge is 1:1.13. The coal seams are well developed and contain 1–4 layers of clips. The clips are primarily thin-bedded mudstones with a uniform thickness. The average dip angle of the coal seam is 10° but falls within the range of 8–14°. The exploited coal seam belongs to the type II spontaneous combustion coal seam. The spontaneous combustion period is 21–55 days, and the coal dust explosion index is 45.79%. At this magnitude, the index qualifies as explosive.

### 2.2 Field monitoring of the three zones of spontaneous combustion in the goaf area

According to the basic characteristics of the 1303 fully mechanized caving faces, the tube monitoring system is placed in the intake and return air lanes. The GC-2010 gas chromatograph

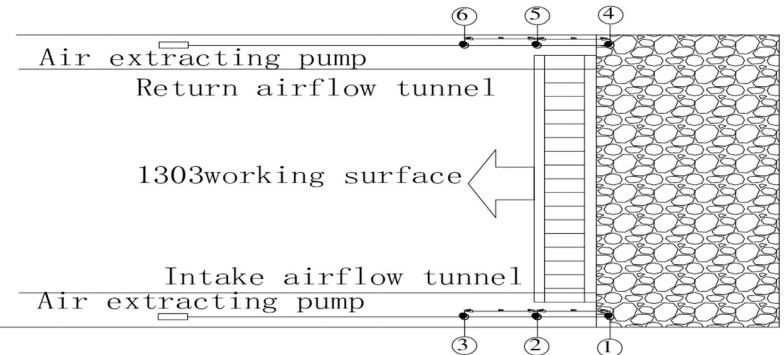

**Fig 1. Layout of beam tube and temperature sensor.**

provided by Jinniu coal mine is used to measure the content of the gas in the goaf. Measuring points are arranged at the same relative positions in the intake and return air tunnels. The mutual distance between the measuring points is 20 m, and there are three measuring points on each side. The specific layout scheme is shown in Fig 1.

The collapse of the roof of the goaf could easily damage the probe at the measuring point. A seamless steel tube is used to protect the beam tube from this damage after probe installation. The tube cable and the temperature measuring wire are fixed into the casing together. The length of the temperature wire is set to prevent breakage caused by tension. The sleeves are connected via a quick joint. A temperature sensor and a sampling head are installed at each measuring point. The sampling head is attached to the beam tube, and the temperature sensor is connected to the temperature measuring wire.

Over the course of two months, the temperature and gas concentrations in the goaf of 1303 fully mechanized caving faces was measured. The dependency of oxygen concentration with respect to the advancing degree of the working face is shown in Fig 2. Measurement point six is not monitored because the beam tube was blocked. Because the air leakage velocity is a vector, field measurement is very difficult. Therefore, it adopts the standard [13–16] of 8–18%,

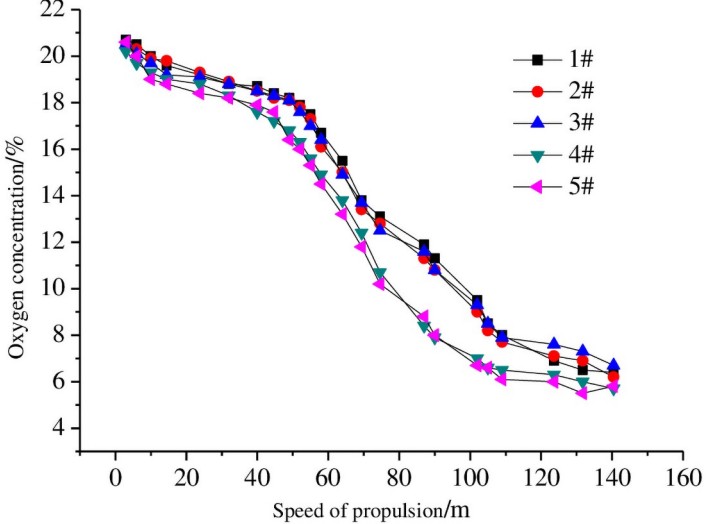

**Fig 2. Change curves of oxygen concentration with working face advancing.**

which is currently used in China. The spontaneous combustion zone of the intake air side is 52–109 m, and the air backside spontaneous combustion zone is 40–92 m.

## 3 Calculation model and solution conditions

### 3.1 Geometric model

According to the field measurements from the 1303 fully mechanized coal caving faces in the Jinniu coal mine, the height of the goaf is far less than the plane size. The two-dimensional model is superior to the three-dimensional model with regard to the accuracy and calculation time. As such, the two-dimensional geometric model of the goaf is shown in Fig 3.

- The length of the working face is 90 m, and the width is 7 m.

- The length of the goaf length GL is dynamically changed, and the width is 90 m.

- The boundary, W1, is the air inlet of the air entry lane with a width of 3 m.

- W2 is the return air inlet of the return air lane with a width of 3 m.

- W3, W4, and W7 are the non-acquired solid coal walls in the working face.

- W5 is the outside protection coal wall of the inlet lane.

- W6 is the outer protective coal wall of the return air lane.

- G1 is the boundary of the goaf of the return air lane.

- G2 is the boundary of the goaf of the intake airflow tunnel

- G3 of the roof pressure stability area is used to calculate the edge of the area.

### 3.2 Mathematical model

When a gas is flowing in the deep part of the goaf at a low velocity and is assumed to be incompressible, fluid dynamics can be characterized as steady-state two-dimensional laminar flow. For this case, energy exchange is not considered. Therefore, the flow of air in the goaf is only needed to meet the momentum conservation equation and the continuity equation [17]. Assuming that the goaf is fully caving and that the top and bottom are not gas-permeable, the following control equation can be obtained:

$$\frac{\partial u}{\partial x} + \frac{\partial v}{\partial y} = 0, \tag{1}$$

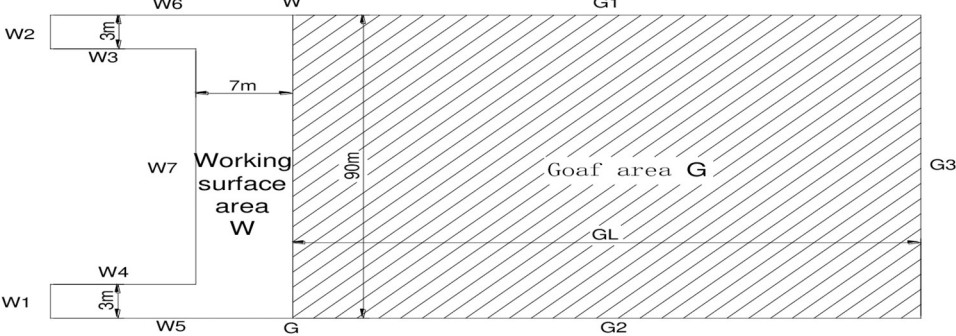

**Fig 3. Geometric model of goaf.**

$$\frac{\partial(\rho u)}{\partial t} + \frac{\partial(\rho uu)}{\partial x} + \frac{\partial(\rho uv)}{\partial y} = \frac{\partial}{\partial x}\left(\mu\frac{\partial u}{\partial x}\right) + \frac{\partial}{\partial y}\left(\mu\frac{\partial u}{\partial y}\right) - \frac{\partial P}{\partial x} + S_x, \tag{2}$$

$$\frac{\partial(\rho v)}{\partial t} + \frac{\partial(\rho uv)}{\partial x} + \frac{\partial(\rho vv)}{\partial y} = \frac{\partial}{\partial x}\left(\mu\frac{\partial u}{\partial x}\right) + \frac{\partial}{\partial y}\left(\mu\frac{\partial u}{\partial y}\right) - \frac{\partial P}{\partial y} + S_y. \tag{3}$$

In the above equations, $u$ and $v$ are velocity (m/s) components in the directions of $x$ and $y$, respectively; $\rho$ is the density (kg/m$^3$) of the air in the mine, $t$ (s) is gas flow time; $P$ is the pressure (Pa) on the fluid microelement; $\mu$ is the air viscosity coefficient (kg·m$^{-1}$·s$^{-1}$) of the goaf; $S_x$ and $S_y$ are model related source items.

The source terms of the porous medium model include two parts: the viscous loss term and the momentum loss term [18]. When the fluid moves in porous media, the loss of momentum causes the pressure gradient to decrease. The pressure gradient drop is proportional to the rate of fluid flow. In a simple homogeneous porous medium:

$$S_i = \frac{\mu}{\alpha_0}v_i + \beta\frac{1}{2}\rho v_j v_j, \tag{4}$$

$$\frac{\partial}{\partial x}\left(k_x\frac{\partial H}{\partial x}\right) + \frac{\partial}{\partial y}\left(k_y\frac{\partial H}{\partial y}\right) = 0, \tag{5}$$

$$Q_x\cdot\frac{dc}{dx} + Q_y\cdot\frac{dc}{dy} = D_x\frac{\partial^2 c}{\partial x^2} + D_y\frac{\partial^2 c}{\partial y^2} - W. \tag{6}$$

In Eqs (4) = (6), $\alpha_0$ is the permeability coefficient of the porous medium; $\beta$ is the coefficient of inertia resistance; $H$ is the roof caving height (m); $Q_x$ and $Q_y$ indicate the air leakage intensity (m$^3$/m$^2$·s) in the directions of $x$ and $y$, respectively; $k$ is absolute permeability (m$^2$); $c$ is the mass concentration of oxygen (kg/m$^3$); $D$ is the diffusion coefficient of oxygen in the coal (m$^2$/s); and $W$ is the oxygen consumption remittance of coal (mol·m$^{-3}$·s$^{-1}$).

To make the mathematical model more suitable for simulating experimental conditions, the influence of residual coal oxidation and gas emission on the oxygen concentration in goaf is considered. Below, a model of oxygen consumption remittance in goaf is constructed:

$$W = -\frac{W(O_2)\cdot H_1}{H} + W(c), \tag{7}$$

$$W(O_2) = \frac{1-n}{n}\cdot\frac{c}{c_0}\cdot y_0\cdot e^{b_0 t}, \tag{8}$$

$$W(c) = \frac{n\cdot H\cdot W(CH_4)}{n\cdot H + W(CH_4)}\cdot c, \tag{9}$$

$$H_1 = k_1[M\cdot(1-\alpha_1) + M_1], \tag{10}$$

$$H = \frac{k_p M}{k_p^{(0)} - 1}. \tag{11}$$

In the formula, $W(O_2)$ is the oxidative oxygen consumption of coal ($mol \cdot m^{-3} \cdot s^{-1}$); $W(c)$ is the equivalent oxygen consumption intensity, considering the dilution effect of gas emission in the goaf ($mol \cdot m^{-3} \cdot s^{-1}$); $W(CH_4)$ is the amount of gas emission from the goaf ($0.12 \sim 4.7$ $mol \cdot m^{-3} \cdot s^{-1}$); $\gamma_0$ is the undetermined coefficient of coal oxygen consumption rate ($0.098$ $mol \cdot m^{-3} \cdot s^{-1}$); $c_0$ is the oxygen concentration in the new airflow (21%); $H_1$ is the thickness of coal remains (m); $n$ is the porosity with a value of 0.25; $\alpha_1$ is the recovery rate of the working face; $k_1$ is a loose coefficient with a value of 1.5; $M_1$ is the thickness of the upper non-recoverable coal seam, this was measured to be 0.4 m; $k_p$ is the coefficient of compaction, this was measured to be $1.1 \sim 1.5$; $k_p^{(0)}$ is initial bulging coefficient; and $M$ is the mining height (4.5 m).

### 3.3 Boundary conditions and calculation parameter settings

The calculation conditions and parameters are selected based on the actual conditions in the field. The entrance boundary of the air entry lane is set as the velocity entrance. The airflow enters vertically into the entry lane of the intake airflow tunnel. The nitrogen injection port is set as the velocity entrance with the nitrogen concentration set at 97% with the remainder constituting the oxygen concentration at 3%. The exit boundary of the goaf is set as free export; the boundary between the working surface and the goaf area is set as the internal condition, and the rest of the surface is set as a wall boundary with a high seepage resistance. All the walls are non-slip boundary conditions, and the standard wall function method is applied near the wall of the working surface.

$$\text{Wall boundary} : E = 0. \tag{12}$$

$$\text{Working surface boundary} : E = R_1 q^2 (l - d)^2. \tag{13}$$

In the above equations, $E$ is the pressure energy loss of fluid (J); $R_1$ is ventilation resistance ($0.0013$ $N \cdot s^2 \cdot m^{-8}$); $q$ is the volume of air passing through the working surface ($m^3$/min); $l$ is the length of the working face (90 m); and $d$ is the distance from the intake to the inlet side (m).

From experimental measurements, the airflow temperature in the intake airflow tunnel is 18.6°C, and the wind speed at the working surface is 1.62 m/s. As a whole, the average mine air density is 1.225 $kg/m^3$, the air viscosity coefficient at room temperature is $1.7894 \times 10^{-5}$ $kg \cdot m^{-1} \cdot s^{-1}$, and the diffusion coefficient of the gas is $2.88 \times 10^{-5}$ $m^3$/s. The looseness coefficient is set to be 1.5. The porosity of the goaf can be obtained from the empirical formula [19] based on the actual situation of roof falling and bulging.

$$\begin{cases} n = 0.00001x^2 - 0.002x + 0.3 (x \leq 100) \\ n = 0.2 (x > 100) \end{cases}. \tag{14}$$

The permeability of the goaf is calculated by the porous medium Carman formula [20]:

$$k = \frac{D_m^2 n^3}{180(1 - n^2)}. \tag{15}$$

In the formula: $x$ is the distance from the working surface of the goaf (m). $D_m$ is the average particle size (m).

## 4 Determination of nitrogen injection amount in the goaf

Before the numerical simulation is conducted, the required nitrogen injection amount is determined according to the actual production conditions and previous experience of the 1303 fully mechanized caving faces. As such, this method calculates the required amount of nitrogen

injection based on the oxygen concentration in the oxidized spontaneous combustion zone. The nitrogen dilutes the original oxygen concentration in the oxidation temperature rising zone to meet the inserting requirement for preventing formation of an oxidation spontaneous combustion zone in the goaf. The amount of nitrogen injection can be calculated by the following formula:

$$Q_N = 60 Q_0 k' \frac{C_1 - C_2}{C_N + C_2 - 1},\qquad(16)$$

where $Q_N$ is the rate of nitrogen injection (m$^3$/h), and $k'$ is an additional coefficient with a value of 1.3. $Q_0$ is the amount of air leakage in the oxidizing rising temperature zone. This was measured to be 3.66 m$^3$/h. $C_1$ is the average oxygen concentration (15%) in the oxidizing rising temperature zone, and $C_2$ is the concentration of oxygen (8%) when the goaf reaches the fire protection requirement. $C_N$ is the concentration of nitrogen injection. As stated earlier, this was set to 97%. From Eq (16) the nitrogen injection amount was calculated to be 400 m$^3$/h.

## 5 Numerical simulation results and analysis

### 5.1 Numerical simulation results

The range of nitrogen diffusion and migration in the goaf will change because of the different positions of nitrogen injection, resulting in changes in the distribution area of the "three-zone" in the goaf. To determine the best position of nitrogen injection, with the nitrogen injection rate defined previously, the distribution of the "three-zone" of spontaneous combustion in the goaf is numerically simulated under the conditions of 10, 20, 30, 40, 50, 60, and 70 m spacing between the position of nitrogen injection and the cutoff line. The model can determine the two classification indices of the flow velocity and oxygen concentration in the comprehensive goaf. The modelled three-band distribution matches the experimental results. Specifically, the isolating lines of wind speed (0.004 m/s) and oxygen concentration (8%) are determined as the upper and lower limits of the oxidation temperature rising zone.

The distribution range of the "three-zone" of spontaneous combustion in goaf under different nitrogen injection positions is obtained by numerical simulation as shown in Fig 4. The red line indicates the equivalent line of the air leakage rate of 0.004 m/s, and the black line indicates the equivalent line of the oxygen concentration of 8%. The figure shows how a change in the location of nitrogen injection results in a different distribution trend for the oxidized spontaneous combustion zone of the goaf. With the movement of nitrogen injection into the deep goaf, the main influence is the 8% oxygen concentration equivalent line, while the variation trend of the 0.004 m/s air leakage velocity is not obvious.

### 5.2 Analysis of numerical simulation results

The width of the oxidized spontaneous combustion zone in the goaf from different nitrogen injection positions can be obtained from Fig 4 and is shown in Table 1. The upper limit of the oxidized spontaneous combustion zone is not significantly affected by the position of the nitrogen injection port, but the position of the nitrogen injection port does influence the lower boundary limit of the oxidized spontaneous combustion zone. When the location of the nitrogen injection moves away from the cutting line, the width of the oxidized spontaneous combustion zone in the goaf begins to decrease followed by a gradual increase. For distances between the injection port and the cutting line of 10, 20, 30, 40, 50, 60, and 70 m, the width of the corresponding oxidized spontaneous combustion zone is 44, 35, 32, 28, 26, 28, and 30.6 m, respectively. The width of the oxidized spontaneous combustion zone is reduced to 75%, 60%,

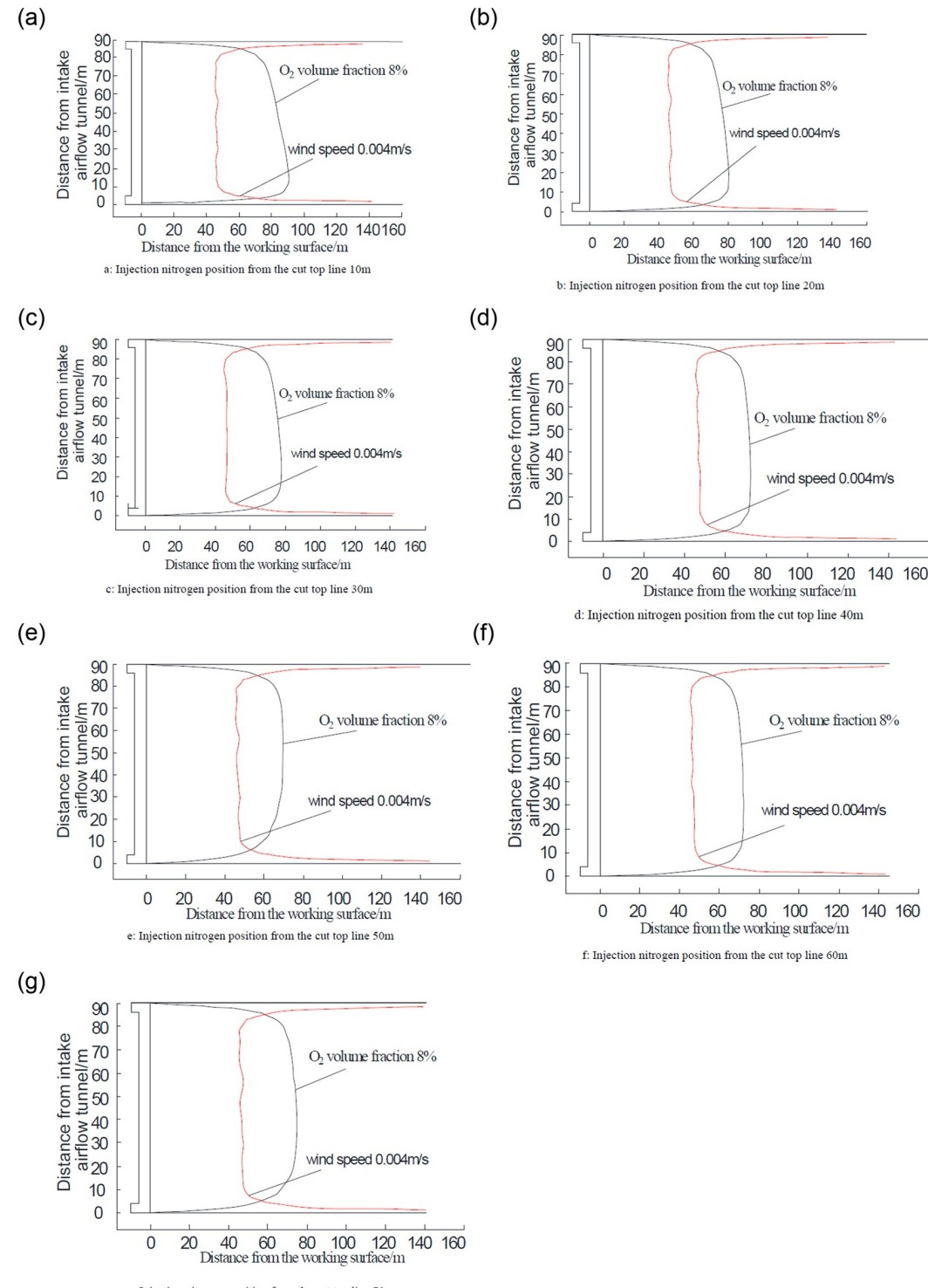

**Fig 4. Goaf oxidized zone distribution under different injection nitrogen position.**

**Table 1. Distribution of oxidized spontaneous combustion zone in goaf with different nitrogen injection position.**

| Position of nitrogen injection (Distance from crest line) /m | Starting position of oxidation zone /m | Termination position of oxidation zone /m | width /m |
| --- | --- | --- | --- |
| 10 | 46.5 | 90.5 | 44 |
| 20 | 46 | 81 | 35 |
| 30 | 45.5 | 77.5 | 32 |
| 40 | 44 | 72 | 28 |
| 50 | 44 | 70 | 26 |
| 60 | 44.5 | 72.5 | 28 |
| 70 | 46 | 76.6 | 30.6 |

55%, 48%, 45%, 48%, and 53%, respectively, under the condition of non-nitrogen injection. This demonstrates that nitrogen injection in the goaf can effectively reduce the width of the oxidized spontaneous combustion zone.

Because the location parameters of nitrogen injection will affect both the impact of the nitrogen injection and operation costs, the model optimizes the nitrogen injection amount based on actual production conditions. From Table 1, the relationship between the width of the oxidized spontaneous combustion zone and the position of the nitrogen injection port is plotted using the Origin software, as shown in Fig 5. As seen in the diagram, the width of the oxidized spontaneous combustion zone gradually diminishes as the position of the nitrogen injection hole becomes deeper and deeper in the goaf. However, the width of the oxidized spontaneous combustion zone begins to increase when the spacing of the nitrogen injection port and the cut top line exceeds 50 m. This occurs because the nitrogen injection port is close to the asphyxiated zone in the goaf when it is buried too deep into the goaf. The oxygen concentration in the shallow positions of the goaf cannot present a proper dilution inserting effect, and the nitrogen injection pressure has minimal influence on the effect of nitrogen injection. As a result, the optimal nitrogen injection position of the goaf should be 40–50 m of the distance working face and the most suitable nitrogen injection position is set as 40 m.

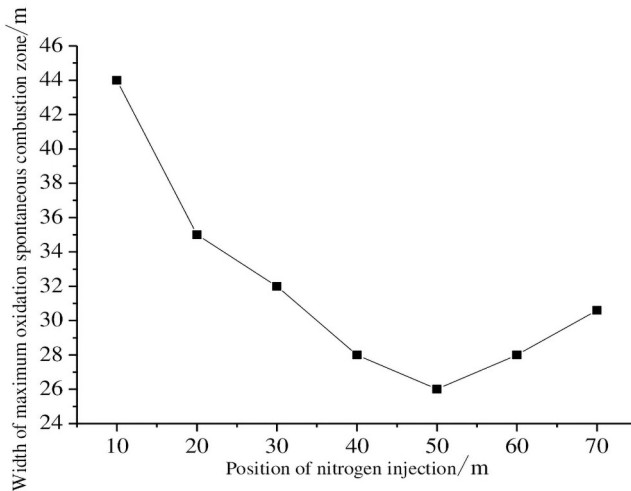

**Fig 5. Goaf oxidized zone width with injection nitrogen position change.**

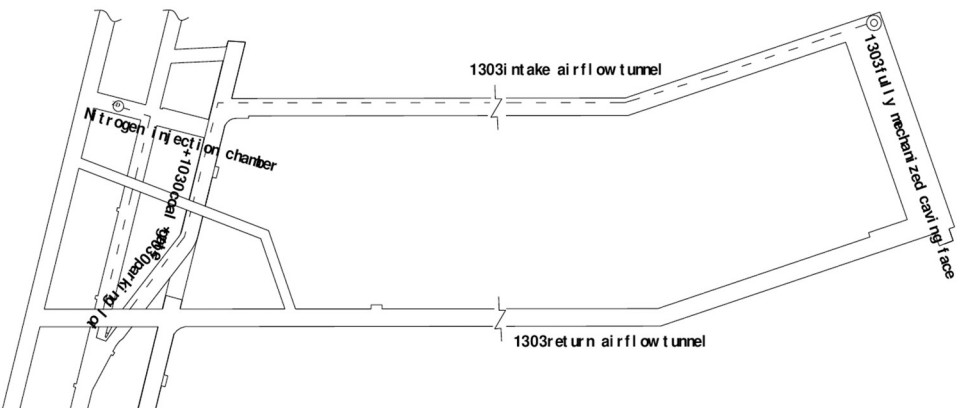

**Fig 6. 1303 fully mechanized top-coal caving face goaf nitrogen injection circuit.**

## 6 Application and effect analysis of nitrogen injection control measures

After consultation with the mineowner, the nitrogen injection pipeline is embedded inside the goaf of the 1303 fully mechanized caving faces of the Jinniu Mine based on the numerical simulation results. The distance between the nitrogen injection port and the top cut line is 40 m, and the nitrogen injection amount is 400 $m^3$/h. The nitrogen source equipment uses the underground mobile molecular sieve nitrogen device (model number DT-400). A 4-inch iron pipe is selected as the nitrogen injection pipeline. The nitrogen injection method is primarily for nitrogen injection in the embedded pipe, and the nitrogen injection line is shown in Fig 6.

After 30 days of nitrogen injection, the gas in the goaf was collected by the existing beam tube monitoring system, and the effect of nitrogen injection on gas concentration in the goaf was analyzed. The impact of the nitrogen injection on the goaf oxygen concentration is shown in Fig 7. From Fig 7, the width of the oxidized spontaneous combustion zone in the goaf is reduced before nitrogen injection. Specifically, the width of the oxidized spontaneous

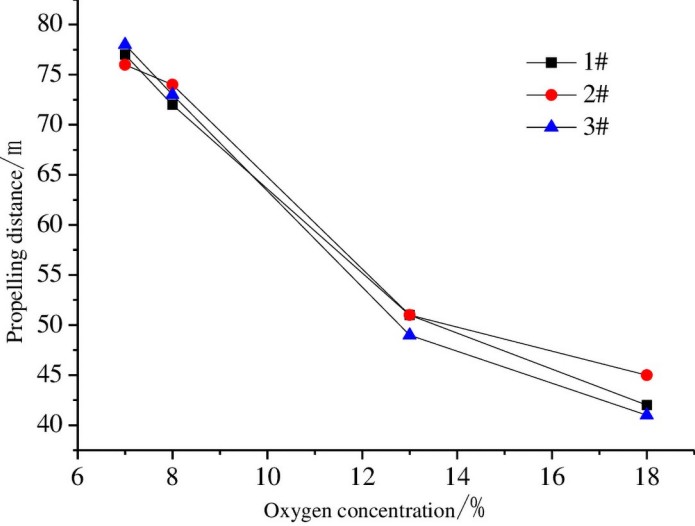

**Fig 7. Change of oxygen concentration at each measuring point.**

combustion zone of the #1 point is 30 m, the width of the #2 point is 29 m, and the width of the #3 point is 32 m. These experimental results are consistent with the maximum width (28 m) of the simulated oxidized spontaneous combustion zone, which verifies the reliability of the numerical simulation results.

To further verify the combustion suppression effect of nitrogen injection on the goaf, the method of combining two indexes of air leakage velocity and oxygen concentration is used to divide the three zones in the goaf after nitrogen injection. Because leakage velocity cannot be measured experimentally, the following formula is used to estimate air leakage speed:

$$Q(d_i) = \frac{v_{o_2}(T)(d_{i+1} - d_i)}{c_{o_2} \ln\left(\frac{c_{o_2}^i}{c_{o_2}^{i+2}}\right)}. \tag{17}$$

In the formula, $Q(d_i)$ is the speed $(cm^3 \cdot s^{-1} \cdot cm^{-2})$ of air leakage at $d_i$, the distance (m) of point $i$ of the goaf from the working face. Also, $v_{o_2}(T)$ is the oxygen consumption rate $(mol \cdot s^{-1} \cdot cm^{-3})$ at a coal body temperature (K) of T. Next, $C_{o_2}$ is the oxygen concentration $(mol/cm^3)$ in the airflow of the intake tunnel. The air leakage velocity calculated by the measured goaf oxygen concentration and formula (15) are used to divide the three zones distribution of spontaneous combustion. The width of the oxidized spontaneous combustion zone of the air inlet is 33 m, and the width of the oxidized spontaneous combustion zone of the return air side is 27 m. The three zones of spontaneous combustion divided by the oxygen concentration and the numerical calculation of the air leakage velocity are in agreement with the numerical simulation. This shows the optimum nitrogen injection parameters determined by the numerical simulation can meet the requirements of the mining industry.

In the context of prevention and control of spontaneous combustion in goaf, there is a minimum value for the recovery rate of the working face. The oxidized spontaneous combustion zone of the goaf is wider before the nitrogen injection measures are taken, and the actual recovery speed of 2.5 m/d cannot ensure the safe recovery of the working face. The parameters of nitrogen injection are optimized by numerical simulation, and the width of the oxidized spontaneous combustion zone is shortened significantly. The width of the oxidized spontaneous combustion zone of the air inlet is shortened by 42%, while the width of the oxidized spontaneous combustion zone of the return air side is shortened by 48%. These conditions satisfy the safe recovery speed,

$$v\tau \geq L. \tag{18}$$

In the formula, $v$ is the recovery speed (m/d) of the working face. In addition, $\tau$ is the natural ignition period (days), and $L$ is the width (m) of the oxidized spontaneous combustion zone. The minimum recovery rate is 1.6 m/d with the use of nitrogen injection, so the working face under the current mining speed is in a safe state. As of January 2018, the 1303 fully mechanized coal caving faces had been successfully pushed forward for over 600 meters, and no spontaneous combustion of coal occurred in the goaf. These results highlight the efficacy of the nitrogen injection control measures for the spontaneous combustion of residual coal in the goaf of 1303 fully mechanized coal caving faces.

## 7 Conclusion

By measuring oxygen concentrations at different points along the working surface, the three-zone area of spontaneous combustion in the goaf area of the 1303 fully mechanized coal caving faces of the Jinniu coal mine was divided. The distance from the working face of the goaf to the

working face (52–109 m) and the range of the return side (40–92 m) from the working face defines the oxidation spontaneous combustion zone.

The COMSOL Multiphysics 5.3 software was used to simulate the change law of the three-zone distribution of spontaneous combustion in the goaf under nitrogen injection. The model found that the upper limit of the oxidized spontaneous combustion zone is not significantly affected by the position of nitrogen injection, as the position of nitrogen injection position moves to the deep goaf. In addition, the model found the lower boundary limit of the oxidized spontaneous combustion zone is significantly affected by the position of nitrogen injection. Finally, the width of the oxidized spontaneous combustion zone begins to shrink followed by a gradual increase.

Combining the actual production status of the working face and the simulation results, nitrogen injection position parameters were optimized. When the nitrogen injection flow rate was set to 400 $m^3$/h and the nitrogen injection port was appropriately extended to the rear of the cutting line (40 m), oxygen was properly diluted to prevent spontaneous combustion, and inerting goaf spontaneous combustion effect is the best, and the width of the oxidized spontaneous combustion zone is shortened by about 30 m.

Based on the results of the simulation analysis, nitrogen injection measures were taken in the goaf of the 1303 fully mechanized caving faces. The experimental results demonstrated that spontaneous combustion of the remaining coal in the goaf has been effectively prevented.

## Supporting information

**S1 File.**
(DOCX)

**S2 File.**
(DOCX)

## Author Contributions

**Data curation:** Yun Qi, Wei Wang.

**Formal analysis:** Zhangxuan Ning, Youli Yao.

**Funding acquisition:** Qingjie Qi.

**Investigation:** Qingjie Qi.

**Methodology:** Yun Qi, Qingjie Qi.

**Project administration:** Qingjie Qi.

**Resources:** Wei Wang, Qingjie Qi.

**Software:** Yun Qi, Wei Wang.

**Supervision:** Zhangxuan Ning, Youli Yao.

**Writing – original draft:** Yun Qi.

**Writing – review & editing:** Yun Qi, Wei Wang.

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
