## [Decision Letter · Decision Letter 0]

22 Jul 2021

PONE-D-21-17754

Distribution of Spontaneous Combustion three zones and Optimization of Nitrogen Injection Location in the Goaf of a fully Mechanized top Coal Caving Face

PLOS ONE

Dear Dr. qi,

Thank you for submitting your manuscript to PLOS ONE. After careful consideration, we feel that it has merit but does not fully meet PLOS ONE’s publication criteria as it currently stands. Therefore, we invite you to submit a revised version of the manuscript that addresses the points raised during the review process.

We look forward to receiving your revised manuscript.

Kind regards,

Dragan Pamucar

Academic Editor

PLOS ONE

Journal Requirements:

"The National Natural Science Foundation of China (Grant: 51274113);

Key special-funded projects of the State key R & D program (2018YFC0807900)"

"Qi Qingjie

 YES"

4. Please amend the manuscript submission data (via Edit Submission) to include authors QI Qingjie, NING Zhangxuan, YAO Youli.

5. Please include your tables as part of your main manuscript and remove the individual files. Please note that supplementary tables should remain uploaded as separate "supporting information" files.

Reviewers' comments:

Reviewer's Responses to Questions

**Comments to the Author**

1. Is the manuscript technically sound, and do the data support the conclusions?

Reviewer #1: Yes

Reviewer #2: Yes

2. Has the statistical analysis been performed appropriately and rigorously? 

Reviewer #1: Yes

Reviewer #2: Yes

3. Have the authors made all data underlying the findings in their manuscript fully available?

Reviewer #1: Yes

Reviewer #2: Yes

4. Is the manuscript presented in an intelligible fashion and written in standard English?

Reviewer #1: Yes

Reviewer #2: Yes

5. Review Comments to the Author

Reviewer #1: Dear editor and author,

Through reading " Distribution of Spontaneous Combustion three zones and Optimization of Nitrogen Injection Location in the Goaf of a fully Mechanized top Coal Caving Face ", it is considered that aiming at the problem of nitrogen waste caused by experience setting of nitrogen injection position in goaf residual coal spontaneous combustion, this paper uses COMSOL multiphysics 5.3 software to study the distribution law of oxidation spontaneous combustion zone in goaf under different nitrogen injection position, which has certain engineering application value.

However, there are still some problems to be discussed:

1.What is the innovation and contribution of the thesis work? Innovation is the basic condition of thesis recruitment and publication.

2.It is mentioned in this paper that two-dimensional model should be used, why not use three-dimensional model? What are the advantages of a two-dimensional model?

3. What is the basis for determining the upper and lower limits of the oxidation temperature rise zone by the isoline of 0.004 m·s-1 wind speed and 8% oxygen concentration in the numerical simulation?

4. The actual recovery speed of 2.5 m/d cannot ensure the safe recovery of the working face, using nitrogen injection can solve this problem. Why not consider other solutions, such as increasing recovery speed?

5. Due to the compressibility of goaf fluid, there will be energy exchange in the process of fluid compression, which leads to the fluid flow can not meet the momentum conservation equation and continuity equation. Is the compressibility of fluid considered in this paper?

6.The three zone distribution of spontaneous combustion in goaf mentioned in this paper is only applied to Jinniu coal mine. Are the results and data universal and suitable for popularization and application?

7.Is the result of field engineering test scientific? Whether it is consistent with practice or supported by other literature.

8.Most of the references in this paper are domestic literature, and the papers of foreign journals are rarely cited.

Reviewer #2: The wide application of fully mechanized top coal caving technology improves the production efficiency of the mine and the advancing speed of the working face. Still, it also leads to increased air leakage intensity in the goaf, more residual coal, and other problems. It increases the probability of spontaneous combustion of residual coal in the goaf. Therefore, in this paper, the research on nitrogen injection technology for fire prevention and extinguishing in the goaf has particular engineering application value.

After modification, the paper is acceptable, and the detailed comments are as follows.

1. Why is it proposed in 1.2 goaf spontaneous combustion "three zones" field monitoring. However, in the numerical simulation results of 4.1, The lower limit of the oxidation temperature rise zone is determined by the contour of the wind speed of 0.004 m/s, not the oxygen concentration. The authors did not analyze the application differences between field monitoring and numerical simulation.

2. Through comparative analysis, it is concluded that the field measurement of the maximum width of the oxidation spontaneous combustion zone is consistent with the numerical simulation results. The authors mentioned that "1303 fully mechanized top coal caving face has been successfully pushed forward for more than 600 meters, and there is no spontaneous combustion of residual coal in the goaf", but should analyze the effect of nitrogen injection in more detail. Please explain how the paper highlights the novelty and practicality of the research results.

3. In the numerical simulation of the distribution of spontaneous combustion "three zones" in goaf, please explain the source of the simplified model and the scientific basis for the simplified process. If so, please describe and quote relevant literature.

4. Explain the data source of Figures 2, 5, and 7, please. Where are the data sources of figures 2, 5, and 7?

5. Please explain why the width of the three belts on the inlet side is more significant than that on the return side.

6. Whether the advancing speed of the working face has an impact on the three zones and whether the author considers it or not.

7. In this paper, the measured temperature of the air inlet is 18.6 ℃. The temperature has a significant influence on the flow process of underground fluid. Is there any influence on the spontaneous combustion of three zones of goaf when the temperature changes in the numerical simulation?

8. There is almost no reference to foreign-related literature in the paper. Is there no related research in foreign journals? The authors should add foreign scholars appropriately. It is better to add the relevant research results in recent years.

6. PLOS authors have the option to publish the peer review history of their article (what does this mean?). If published, this will include your full peer review and any attached files.

Reviewer #1: No

Reviewer #2: No

---

## [Author Response · Author response to Decision Letter 0]

4 Aug 2021

Dear editor and reviewers:

Thank you for your letter and the reviewers’ comments on our manuscript entitled " Distribution of Spontaneous Combustion three zones and Optimization of Nitrogen Injection Location in the Goaf of a fully Mechanized top Coal Caving Face". Those comments are very helpful for revising and improving our paper, as well as the important guiding significance to other research. We have studied the comments carefully and made corrections which we hope meet with approval. The main corrections are in the manuscript and the responds to the reviewers’ comments are as follows (the replies are highlighted in blue ).

Replies to the reviewers’ comments:

Response to reviewers 1

1.What is the innovation and contribution of the thesis work? Innovation is the basic condition of thesis recruitment and publication.

Response: Predecessors have made detailed research on the principle of nitrogen injection fire prevention and extinguishing and nitrogen injection technology, but they did not consider the influence of residual coal oxidation and gas emission in goaf. In this paper, experimental research is carried out in 1303 fully mechanized top coal caving face of Jinniu coal mine, field measurement is carried out, and the distribution area of spontaneous combustion "three zones" in goaf is divided. COMSOL multiphysics 5.3 software is used to simulate the change of distribution area of spontaneous combustion "three zones" in goaf with the position of nitrogen injection port by coupling air leakage flow field and oxygen concentration field in goaf. The influence of nitrogen injection port position on the width of oxidation spontaneous combustion zone is analyzed, and the most suitable nitrogen injection position is optimized. According to the simulation analysis results, guiding the implementation of on-site nitrogen injection scheme is of great significance to change the previous nitrogen injection position and nitrogen injection amount, which is set only by experience, resulting in nitrogen waste.

2.It is mentioned in this paper that two-dimensional model should be used, why not use three-dimensional model? What are the advantages of a two-dimensional model?

Response: According to the data of 1303 fully mechanized top coal caving face in Jinniu coal mine and the field measured results, it can be seen that the height of goaf is far less than the plane size, and the two-dimensional model is better than the three-dimensional model in accuracy and calculation time. The results calculated by the two-dimensional model are more intuitive and more targeted. Although the three-dimensional model is closer to the real situation of the mine, the calculation process is too complex and there are many influencing factors to be considered. Therefore, the goaf model established in this paper is a two-dimensional geometric model.

3. What is the basis for determining the upper and lower limits of the oxidation temperature rise zone by the isoline of 0.004 m·s-1 wind speed and 8% oxygen concentration in the numerical simulation?

Response: In the on-site monitoring of the "three zones" of spontaneous combustion in goaf, it is proposed that "the division standard is 8% ~ 18% according to the oxygen concentration" and "the lower limit of oxidation temperature rise zone is determined by the wind speed 0.004 m / s isoline rather than the oxygen concentration" in the numerical simulation results in 4.1. According to the division of the "three zones" of spontaneous combustion in goaf, the oxygen concentration division method is widely used in China Air leakage wind speed division method (numerical simulation method) and heating rate division method. The indexes of oxidation temperature rise zone in goaf corresponding to the three division methods are the areas with oxygen concentration of 8% ~ 18%, air leakage wind speed of 0.0016 ~ 0.004m/s and temperature rise rate k > 10 ℃ / D in goaf.

The reason why the division standard of oxygen concentration of 8% ~ 18% is adopted in the actual measurement is to follow the oxygen concentration index widely used in the field measurement of three zones in goaf in China. In the numerical simulation, the wind speed of 0.004 m /s is adopted because the air leakage velocity in the goaf is a vector, which is difficult to measure in practice. It is usually used to establish and solve the mathematical model. The streamline and wind speed distribution of air leakage in the goaf under different boundary conditions can only be numerically simulated by computer. In view of the large amount of gas emission in the goaf of 1303 fully mechanized top coal caving face of Jinniu coal mine, The author believes that the combination of oxygen concentration and air leakage wind speed is more reasonable to divide the "three zones" of spontaneous combustion in goaf. That is, the boundary between the diffuse zone and the oxidation zone is limited to the wind speed of 0.004m/s, and the boundary between the oxidation zone and the asphyxia zone is limited to the oxygen concentration of 8%.

4. The actual recovery speed of 2.5 m/d cannot ensure the safe recovery of the working face, using nitrogen injection can solve this problem. Why not consider other solutions, such as increasing recovery speed?

Response: The mining speed of each coal mine is determined by underground geological conditions, parameters of mining machinery and equipment, effective operation time and other factors. Once the coal mining speed is determined, it can not be changed at will. At present, the coal mining speed of Jinniu coal mine is 2.5m/d, which can not ensure the safe mining of coal mining. Therefore, corresponding measures need to be taken to solve the problem of spontaneous combustion in goaf under this speed. Because the nitrogen injection fire prevention and extinguishing process is simple, easy to operate and does not pollute the fire prevention and extinguishing area, in addition, the nitrogen injection fire prevention and extinguishing has little damage to the equipment in the closed area, can effectively suppress the air leakage in the fire prevention and extinguishing area, and has good dilution and explosion suppression performance. If we blindly increase the mining speed, it will only increase the air leakage intensity and the risk of spontaneous combustion in the goaf. Therefore, nitrogen injection is selected as a fire prevention and extinguishing measure in this paper.

5. Due to the compressibility of goaf fluid, there will be energy exchange in the process of fluid compression, which leads to the fluid flow can not meet the momentum conservation equation and continuity equation. Is the compressibility of fluid considered in this paper?

Response: The problem of fluid compressibility is not considered in this paper. There will be energy changes after fluid compression, but because the fluid flow velocity in the goaf is relatively small, the gas in the goaf is assumed to be incompressible gas in this paper, and the energy exchange caused by gas compression is not considered. Therefore, the air flow in the goaf can meet the momentum conservation equation and continuity equation. Fluid compressibility will be considered in future research.

6. The three zone distribution of spontaneous combustion in goaf mentioned in this paper is only applied to Jinniu coal mine. Are the results and data universal and suitable for popularization and application?

Response: The distribution characteristics of three spontaneous combustion zones in goaf are universal, but each coal mine will lead to different positions and widths of three spontaneous combustion zones in goaf due to different geological factors, coal seam burial depth and coal seam thickness. Therefore, the results and data calculated according to the calculation parameters set according to the actual situation of Jinniu coal mine are not universal, However, the distribution characteristics of the three spontaneous combustion zones obtained are universal and can be extended to similar mining areas.

7. Is the result of field engineering test scientific? Whether it is consistent with practice or supported by other literature.

Response:According to the optimization results of nitrogen injection position in the paper, the field engineering test is carried out, and the changes of oxidation spontaneous combustion zone width in goaf before and after nitrogen injection are obtained, which verifies the inerting effect of nitrogen injection on spontaneous combustion risk in goaf, and verifies the correctness and scientificity of numerical simulation. This is consistent with the results in literature [9] -[10].

8. Most of the references in this paper are domestic literature, and the papers of foreign journals are rarely cited.

Response: It has been revised, and the research results of four foreign scholars have been cited in the paper. In addition, most references have been replaced by papers in foreign journals.

Response to reviewers 2

1. Why is it proposed in 1.2 goaf spontaneous combustion "three zones" field monitoring. However, in the numerical simulation results of 4.1, The lower limit of the oxidation temperature rise zone is determined by the contour of the wind speed of 0.004 m/s, not the oxygen concentration. The authors did not analyze the application differences between field monitoring and numerical simulation.

Response: Using the division standard of oxygen concentration of 8% ~ 18% to monitor the three spontaneous combustion zones in goaf is the most commonly used method in China. This method is not only simple, convenient and easy to operate, but also can reflect the oxidation oxygen supply and heat storage conditions of residual coal. However, the standard of wind speed of 0.004 m / s is adopted in the later numerical simulation, mainly because the wind speed is a vector in the actual measurement, and it is difficult to solve it through the mathematical model after the actual measurement. We can only use the computer to simulate the air leakage flow line and wind speed distribution in the goaf under different boundary conditions. In view of the characteristics of goaf spontaneous combustion risk in 1303 fully mechanized top coal caving face of Jinniu coal mine, the author believes that it is more reasonable to divide the "three zones" of goaf spontaneous combustion by using the measured oxygen concentration and numerical simulation with air leakage wind speed.

2. Through comparative analysis, it is concluded that the field measurement of the maximum width of the oxidation spontaneous combustion zone is consistent with the numerical simulation results. The authors mentioned that "1303 fully mechanized top coal caving face has been successfully pushed forward for more than 600 meters, and there is no spontaneous combustion of residual coal in the goaf", but should analyze the effect of nitrogen injection in more detail. Please explain how the paper highlights the novelty and practicality of the research results.

Response: In this paper, the distribution law of oxidation spontaneous combustion zone in goaf under different nitrogen injection port positions is numerically simulated and analyzed, so as to select the best nitrogen injection port position in goaf, determine nitrogen injection parameters by means of numerical simulation, save money and shorten the project cycle; According to the simulated optimal nitrogen injection parameters, the nitrogen injection scheme in goaf of 1303 fully mechanized top coal caving face is formulated and applied in the field. The consistency between simulated nitrogen injection and actual nitrogen injection is verified by the existing goaf riser monitoring system; After nitrogen injection measures are taken in the goaf, the oxidized natural zone is significantly reduced, and the minimum recovery speed is 1.6m/d, which is less than the current actual recovery speed of 2.5m/d. Therefore, it can ensure the safe recovery of the working face. The distribution law of oxidation spontaneous combustion zone in goaf under different nitrogen injection port positions and the practical application effect after the optimization of field nitrogen injection related parameters such as determining the best nitrogen injection port position can be seen in the text below figure 7 in the revised draft.

3. In the numerical simulation of the distribution of spontaneous combustion "three zones" in goaf, please explain the source of the simplified model and the scientific basis for the simplified process. If so, please describe and quote relevant literature.

Response: When the model is established, a large number of references are referred for corresponding simplification, such as references [4], [6], [9]. The simplified model includes wall boundary and working face boundary, which can meet the simulation of three zones of spontaneous combustion in goaf and simplify the calculation process. The calculation results of the simplified model are consistent with the actual distribution of three zones in the goaf of Jinniu coal mine, so the simplified process is scientific and accurate.

4. Explain the data source of Figures 2, 5, and 7, please. Where are the data sources of figures 2, 5, and 7?

Response:The data sources in figure 2, figure 5 and figure 7 have been submitted in the modify description file.

5. Please explain why the width of the three belts on the inlet side is more significant than that on the return side.

Response:When fresh air enters the shaft and roadway from the air inlet side, it will take away the gas in the shaft and roadway and the heat generated by partial oxidation, so that the tropical area on the air inlet side of the goaf has sufficient oxygen supply conditions, while the gas in the air on the return side accumulates, the gas concentration is greater than that on the air inlet side, and the heat accumulation is more, and the risk of spontaneous combustion on the return side increases significantly. Therefore, the gas concentration on the air inlet side of the goaf and the heat storage capacity is smaller than that of the return air side. Therefore, the width of the three spontaneous combustion zones in the goaf on the inlet side is greater than that on the return air side.

6. Whether the advancing speed of the working face has an impact on the three zones and whether the author considers it or not.

Response:Yes, the advancing speed determines the air leakage intensity of the goaf. The greater the air leakage intensity, the greater the risk of spontaneous combustion in the goaf. However, the research content of this paper does not consider the impact of the advancing speed of the working face on the three zones of spontaneous combustion in the goaf. This is the next research work of the paper. In the future, a dynamic numerical model will be established to dynamically simulate the impact of the advancing speed on the three zones of spontaneous combustion.

7. In this paper, the measured temperature of the air inlet is 18.6 ℃. The temperature has a significant influence on the flow process of underground fluid. Is there any influence on the spontaneous combustion of three zones of goaf when the temperature changes in the numerical simulation?

Response:The underground temperature of coal mine changes dynamically with time, but it basically fluctuates up and down at a fixed value. The influence of this fluctuation range on the three zones of spontaneous combustion is basically negligible. Therefore, the influence of temperature change on the three zones of spontaneous combustion in goaf is not considered in this paper, but the temperature will be analyzed as an influencing factor in subsequent research.

8. There is almost no reference to foreign-related literature in the paper. Is there no related research in foreign journals? The authors should add foreign scholars appropriately. It is better to add the relevant research results in recent years.

Response： it has been modified(the replies are highlighted in blue in the paper). 

Once again, thank you very much for your constructive comments and suggestions which would help us both in English and in depth to improve the quality of the paper.

Kind regards,

Yun QI

E-mail: qiyun_sx@sxdtdx.edu.cn，

Corresponding author : Yun QI, Wei WANG

E-mail address: qiyun_sx@sxdtdx.edu.cn，wangwei@sxdtdx.edu.cn .

---

## [Decision Letter · Decision Letter 1]

10 Aug 2021

PONE-D-21-17754R1

Distribution of Spontaneous Combustion three zones and Optimization of Nitrogen Injection Location in the Goaf of a fully Mechanized top Coal Caving Face

PLOS ONE

Dear Dr. qi,

Thank you for submitting your manuscript to PLOS ONE. After careful consideration, we feel that it has merit but does not fully meet PLOS ONE’s publication criteria as it currently stands. Therefore, we invite you to submit a revised version of the manuscript that addresses the points raised during the review process.

We look forward to receiving your revised manuscript.

Kind regards,

Dragan Pamucar

Academic Editor

PLOS ONE

Journal Requirements:

Reviewers' comments:

Reviewer's Responses to Questions

**Comments to the Author**

1. If the authors have adequately addressed your comments raised in a previous round of review and you feel that this manuscript is now acceptable for publication, you may indicate that here to bypass the “Comments to the Author” section, enter your conflict of interest statement in the “Confidential to Editor” section, and submit your "Accept" recommendation.

Reviewer #1: All comments have been addressed

Reviewer #2: (No Response)

2. Is the manuscript technically sound, and do the data support the conclusions?

Reviewer #1: Yes

Reviewer #2: (No Response)

3. Has the statistical analysis been performed appropriately and rigorously? 

Reviewer #1: Yes

Reviewer #2: (No Response)

4. Have the authors made all data underlying the findings in their manuscript fully available?

Reviewer #1: Yes

Reviewer #2: (No Response)

5. Is the manuscript presented in an intelligible fashion and written in standard English?

Reviewer #1: (No Response)

Reviewer #2: (No Response)

6. Review Comments to the Author

Reviewer #1: The capitalization in the title of the thesis is wrong, it is recommended to make the following changes: Distribution of Three Spontaneous Combustion Zones and Optimization of Nitrogen Injection Sites in the Goaf of a Fully Mechanized Carving Face

Reviewer #2: (No Response)

7. PLOS authors have the option to publish the peer review history of their article (what does this mean?). If published, this will include your full peer review and any attached files.

Reviewer #1: No

Reviewer #2: No

---

## [Author Response · Author response to Decision Letter 1]

12 Aug 2021

The original title of the paper was " Distribution of Spontaneous Combustion three zones and Optimization of Nitrogen Injection Location in the Goaf of a fully Mechanized top Coal Caving Face", which has been polished by Letpub mechanism. Therefore, I think the translation of the title of the paper does not need to be modified, but the case format of the title is wrong, which has been modified now

---

## [Decision Letter · Decision Letter 2]

19 Aug 2021

Distribution of Spontaneous Combustion three zones and Optimization of Nitrogen Injection Location in the Goaf of a fully Mechanized top Coal Caving Face

PONE-D-21-17754R2

Dear Dr. qi,

We’re pleased to inform you that your manuscript has been judged scientifically suitable for publication and will be formally accepted for publication once it meets all outstanding technical requirements.

Kind regards,

Dragan Pamucar

Academic Editor

PLOS ONE

Additional Editor Comments (optional):

Reviewers' comments:

Reviewer's Responses to Questions

**Comments to the Author**

1. If the authors have adequately addressed your comments raised in a previous round of review and you feel that this manuscript is now acceptable for publication, you may indicate that here to bypass the “Comments to the Author” section, enter your conflict of interest statement in the “Confidential to Editor” section, and submit your "Accept" recommendation.

Reviewer #1: All comments have been addressed

Reviewer #2: All comments have been addressed

2. Is the manuscript technically sound, and do the data support the conclusions?

Reviewer #1: Yes

Reviewer #2: Yes

3. Has the statistical analysis been performed appropriately and rigorously? 

Reviewer #1: Yes

Reviewer #2: Yes

4. Have the authors made all data underlying the findings in their manuscript fully available?

Reviewer #1: Yes

Reviewer #2: Yes

5. Is the manuscript presented in an intelligible fashion and written in standard English?

Reviewer #1: Yes

Reviewer #2: Yes

6. Review Comments to the Author

Reviewer #1: (No Response)

Reviewer #2: (No Response)

7. PLOS authors have the option to publish the peer review history of their article (what does this mean?). If published, this will include your full peer review and any attached files.

Reviewer #1: No

Reviewer #2: No

---

## [Editor Report · Acceptance letter]

3 Sep 2021

PONE-D-21-17754R2 

Distribution of Spontaneous Combustion Three Zones and Optimization of Nitrogen Injection Location in the Goaf of a Fully Mechanized Top Coal Caving Face 

Dear Dr. Qi:

I'm pleased to inform you that your manuscript has been deemed suitable for publication in PLOS ONE. Congratulations! Your manuscript is now with our production department. 

Kind regards, 

on behalf of

Dr. Dragan Pamucar 

Academic Editor

PLOS ONE